# Apical dehydration impairs the cystic fibrosis airway epithelium barrier via a $\beta$1-integrin/YAP1 pathway

Juliette L Simonin[1], Caterina Tomba[2], Vincent Mercier[2], Marc Bacchetta[1], Tahir Idris[1], Mehdi Badaoui[1], Aurélien Roux[2], Marc Chanson[1]

Defective hydration of airway surface mucosa is associated with lung infection in cystic fibrosis (CF), partly caused by disruption of the epithelial barrier integrity. Although rehydration of the CF airway surface liquid (ASL) alleviates epithelium vulnerability to infection by junctional protein expression, the mechanisms linking ASL to barrier integrity are unknown. We show here the strong degradation of YAP1 and TAZ proteins in well-polarized CF human airway epithelial cells (HAECs), a process that was prevented by ASL rehydration. Conditional silencing of *YAP1* in rehydrated CF HAECs indicated that YAP1 expression was necessary for the maintenance of junctional complexes. A higher plasma membrane tension in CF HAECs reduced endocytosis, concurrent with the maintenance of active $\beta$1-integrin ectopically located at the apical membrane. Pharmacological inhibition of $\beta$1-integrin accumulation restored YAP1 expression in CF HAECs. These results indicate that dehydration of the CF ASL affects epithelial plasma membrane tension, resulting in ectopic activation of a $\beta$1-integrin/YAP1 signaling pathway associated with degradation of junctional proteins.

## Introduction

Integrins mediate bidirectional signals between the ECM and epithelial cells (Manninen, 2015). These receptors are predominantly located in the basal membrane, whereas junctional complexes delimit apical from basolateral membranes. Junctional complexes include the tight junctions and the adherens junctions. Both tight (claudin-mediated) and adherens (cadherin/catenin-mediated) junctions form bonds between membranes of contacting cells, thereby regulating the transepithelial passage of ions and water-soluble molecules (Zihni et al, 2016; Adil et al, 2021). Thus, the mechanical forces, transduced by integrins and junctional complexes that are exerted on epithelial cells, are critical in guiding the establishment of apicobasal polarity (Manninen, 2015).

In humans, the conducting portion of the airways is lined by a pseudostratified epithelium constituted of progenitor basal cells, and luminal secretory and multiciliated cells (Deprez et al, 2020). The structural integrity and stable barrier function of the airway epithelium form the first line of defense against inhaled environmental insults (Whitsett & Alenghat, 2015). The Yes1-associated protein (YAP1) is critical for progenitor cell maintenance and self-renewal, the amount of YAP1 in cells being essential to shape the pseudostratified architecture of the airway epithelium (Gumbiner & Kim, 2014; Mahoney et al, 2014; Zhao et al, 2014). YAP1 is a known effector of the Hippo signaling pathway, which has emerged as a crucial integrator of signals in biological events from development to adulthood (van Soldt & Cardoso, 2020). YAP1 and its co-activator TAZ (transcriptional co-activator with a PDZ-binding motif) act as transcriptional sensors of the structural and mechanical features of the cell microenvironment via ECM stiffness and cell–cell adhesion junctional complexes (Totaro et al, 2018; Dasgupta & McCollum, 2019; van Soldt & Cardoso, 2020). Thus, mechanosensing through Hippo/YAP signaling regulates cell proliferation, differentiation, and organ size. YAP1 is tightly regulated, and its nuclear localization and transcriptional activity are suppressed by the Hippo cascade kinases when the pathway is active (van Soldt & Cardoso, 2020).

Under certain conditions, such as tissue remodeling, wound repair, or pathological states, integrins can redistribute to the apical membrane (Peterson & Koval, 2021). This phenotype is typically observed in cystic fibrosis (CF), a genetic disease caused by dysfunction of the CFTR anion channel. CFTR is expressed at the apical membrane of human airway epithelial cells (HAECs), and its activity is essential to keep the airway surface liquid (ASL) hydrated (Saint-Criq & Gray, 2017). In CF, ASL dehydration is associated with the loss of barrier function and junctional integrity of the airway epithelium, promoting chronic lung infection and inflammation, and eventually leading to respiratory failure (Shteinberg et al, 2021). Relocation of $\beta$1-integrin in the apical membrane of polarized epithelial cells can have significant implications for cell behavior

[1]Department of Cell Physiology and Metabolism, University of Geneva, Faculty of Medicine, Geneva, Switzerland    [2]Department of Biochemistry, Faculty of Sciences, University of Geneva, Geneva, Switzerland

Correspondence: Marc.Chanson@unige.ch
Caterina Tomba's present address is CNRS, INSA Lyon, Ecole Centrale de Lyon, Université Claude Bernard Lyon 1, CPE Lyon, INL, UMR5270, Villeurbanne, France

and tissue organization. Interestingly, the apical expression of β1-integrin in CF HAECs was shown to act as a receptor for *Pseudomonas aeruginosa*, the main morbidity-causing pathogen in this disease (Grassme et al, 2017; Badaoui et al, 2020). Whether apical β1-integrin–mediated intracellular signaling or trafficking may also contribute to the dysfunctional CF airway epithelium barrier is not known.

Regarding the roles of β1-integrin and YAP1 signaling in cell differentiation (van Soldt & Cardoso, 2020), we hypothesize here that the Hippo/YAP1 pathway may contribute to the links between the apical β1-integrin expression and the loss of junctional network integrity in response to ASL dehydration. To this end, we investigated in a *CFTR* knockdown (CFTR-KD) HAEC model (Bellec et al, 2015) the effects of rehydration on β1-integrin localization, YAP1 expression, and its causal relationship with key proteins of tight and adherens junctions.

# Results

## Rehydration of the CFTR-KD Calu-3 cell surface restored apicobasal polarity of the CF airway epithelium

The generation and characterization of the Calu-3 HAEC line knocked down for *CFTR* (CFTR-KD) and its control counterpart (CFTR-CTL) have been previously reported (Bellec et al, 2015; Badaoui et al, 2020, 2023; Simonin et al, 2022). CFTR-CTL Calu-3 cells are known to efficiently produce apical secretions (CTL-ASL) when grown on Transwell filters at an air–liquid interface (ALI); in contrast, CFTR-KD cells produced very low volumes of ASL (Simonin et al, 2022). Under basal conditions, total (Fig 1A) and active (Fig 1B) β1-integrins were detected at the apical membrane of 3-wk-old CFTR-KD ALI cultures as compared to CFTR-CTL cells. Active from total β1-integrin can be distinguished using the 9EG7 antibody, which binds to an epitope accessible only after activation of this integrin (Lenter et al, 1993). To determine a potential relationship between the presence of an ASL and the localization of β1-integrin at the apical membrane, we compared CFTR-CTL and CFTR-KD ALI cultures with CFTR-KD cells after rehydration of their apical surface. To this end, CTL-ASL from 3-wk-old CFTR-CTL cultures was collected and transferred onto the apical side of CFTR-KD cells for 48 h (CFTR-KD + CTL-ASL 48 h). In parallel experiments, a similar volume of physiological saline was added to the surface of CFTR-KD cells (CFTR-KD + Saline 48 h). Rehydration of CFTR-KD cell cultures with either CTL-ASL or saline was associated with the loss of total (Fig 1A) and active (Fig 1B) β1-integrin detection at the apical surface. Quantitation of these experiments is shown in Fig 1C and D for total and active β1-integrin, respectively.

To evaluate the kinetics of β1-integrin's apical signal extinction, we monitored its expression during rehydration with saline by co-immunofluorescence and Western blot as a function of time. As shown in Fig 2A, the immunofluorescent signal for total and active β1-integrins rapidly vanished, being barely detectable within 1 h of rehydration of CFTR-KD cultures. 3D projections of the airway epithelium are shown in Fig S1. However, the total expression of β1-integrin determined by Western blot remained unchanged during

this lap of time (Fig 2B and C). These results suggest that rehydration of the CFTR-KD cell surface restored apicobasal polarity of the CF airway epithelium without affecting total β1-integrin expression.

## Rehydration of the CFTR-KD Calu-3 cell surface increased endocytosis by modulating plasma membrane tension

Because the previous results are consistent with a rapid internalization of the β1-integrin apical membrane pool, we next investigated whether rehydration could affect the endocytic process. Dextran is a commonly used probe to monitor fluid-phase endocytosis (Ramoino et al, 2001). Thus, dextran uptake was evaluated in CFTR-KD cultures and CFTR-KD cells after 1-h rehydration with saline. As shown in Fig 3A, intracellular accumulation of dextran was observed after 1-h rehydration of CFTR-KD cultures, whereas this rehydration did not change cell volume (Fig 3B). To quantify dye internalization, images of dextran, phalloidin, and DAPI fluorescent signals were acquired by Z-stack confocal microscopy of ALI cultures and were reconstructed using Imaris software, as shown in Fig S2A. Each dextran-positive structure was quantified and normalized per cell number (Fig 3C), per cell volume (Fig 3D), or to the apical surface (Fig 3E). Regardless of the quantification type, rehydration increased all measurements in CFTR-KD cultures in agreement with a strong endocytosis activation. The distance of a dextran signal from the phalloidin-stained subcortical actin, which follows the membrane, was also measured. The distribution of the values corresponding to the shortest distance between the two signals is shown in Fig S2B for all acquired images obtained in four separate experiments. As shown in Fig S2C, rehydrated CFTR-KD cultures exhibited a higher ($P = 2.577 \times 10^{-37}$) fraction of median distance to the membrane (0.52 $\mu$m) as compared to control CFTR-KD cells (0.38 $\mu$m). These results indicate that endocytosis is strongly diminished in dehydrated CFTR-KD cells and stimulated with the apical addition of the saline medium, providing an explanation for the maintenance of apical β1-integrin expression.

Altered β1-integrin endocytosis can be due to an increased plasma membrane tension, which is adjusted in response to environmental cues (Lolo et al, 2022). We thus examined whether rehydration of CFTR-KD cultures could modulate their membrane mechanical properties. To address this question, we used the FliptR probe, a fluorescent reporter of membrane tension (Colom et al, 2018), to both CFTR-CTL and CFTR-KD ALI cultures. Using fluorescence lifetime imaging microscopy (FLIM), we found increased FliptR lifetime in CFTR-KD cultures as compared to CFTR-CTL cells (Fig 4A), which can reflect a different lipid composition or a higher membrane tension. In addition, we observed increased lifetime of FliptR after CTL-ASL removal from CFTR-CTL cells (Fig 4B), whereas apical rehydration of CFTR-KD cultures with saline decreased FliptR lifetime within minutes (Fig 4C). The fast kinetics of FliptR lifetime changes with ASL manipulation indicate that membrane tension rather than lipid composition is modulated by the presence of an apical liquid volume. These results suggest that plasma membrane tension is adjusted to the mechanical stress exerted by the ASL, which in turn controls endocytosis.

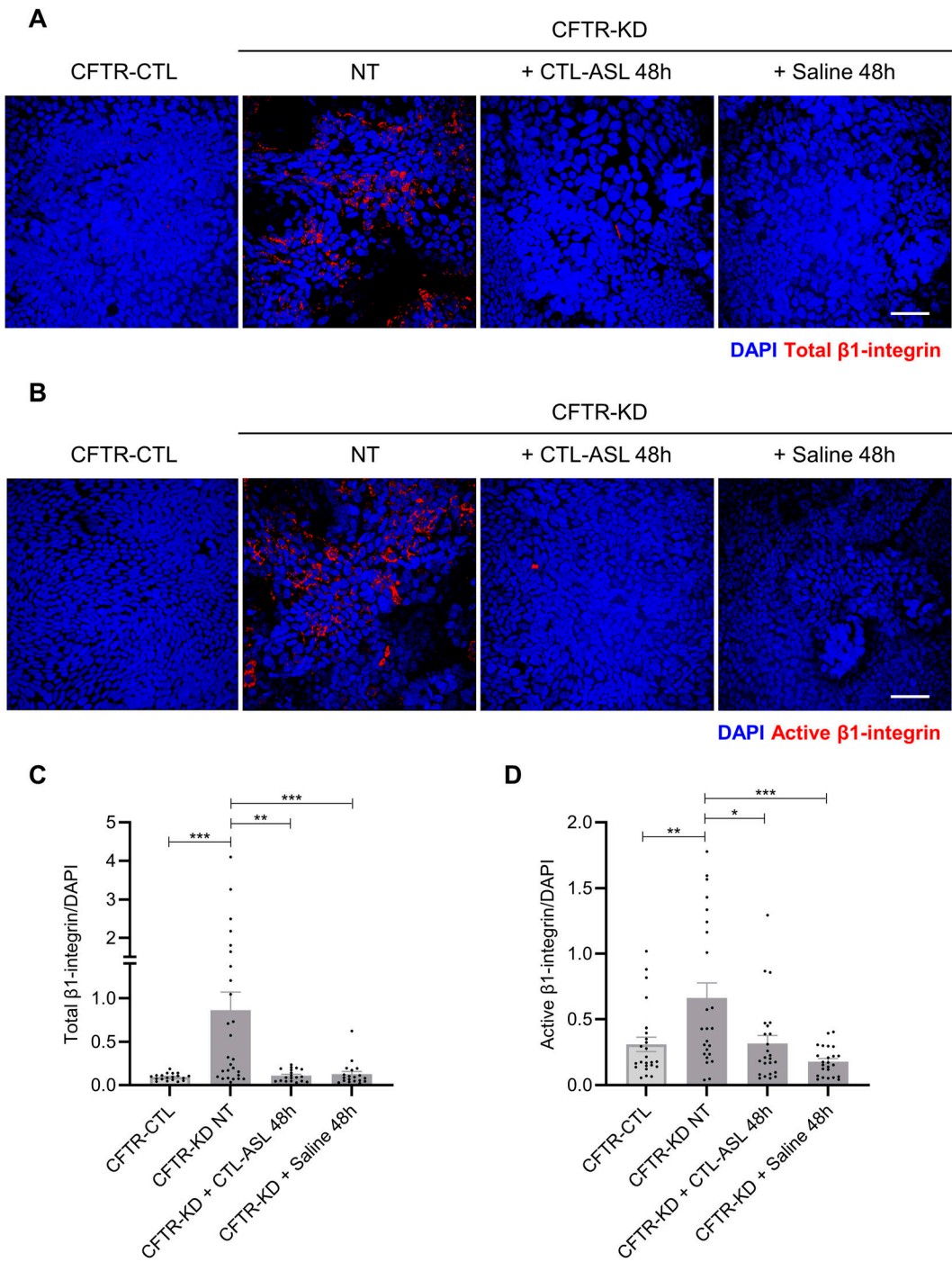

**Figure 1. Rehydration eliminates apical β1-integrin localization at the CFTR-KD cell surface.**
**(A, B)** Localization of total (A) and active (B) β1-integrins was evaluated by immunofluorescence in polarized CFTR-CTL and CFTR-KD ALI cultures, as markers of apicobasal polarity. Representative apical images from z-stack confocal acquisitions are shown in CFTR-CTL and in CFTR-KD cultures before (not treated, NT) and after 48-h rehydration with CTL-ASL or saline. DAPI: blue; β1-integrin: red. Scale bar: 50 μm. **(C, D)** Quantitation of the immunofluorescent signal normalized to the number of cells (DAPI) is shown for (C) total β1-integrin (CFTR-KD NT, $P$ = 0.0003; CFTR-KD + CTL-ASL 48 h, $P$ = 0.0011; CFTR-KD + Saline 48 h, $P$ = 0.0006) and (D) active β1-integrin (CFTR-KD NT, $P$ = 0.0082; CFTR-KD + CTL-ASL 48 h, $P$ = 0.0153; CFTR-KD + Saline 48 h, $P$ < 0.0001) ($N$ = 3).
Source data are available for this figure.

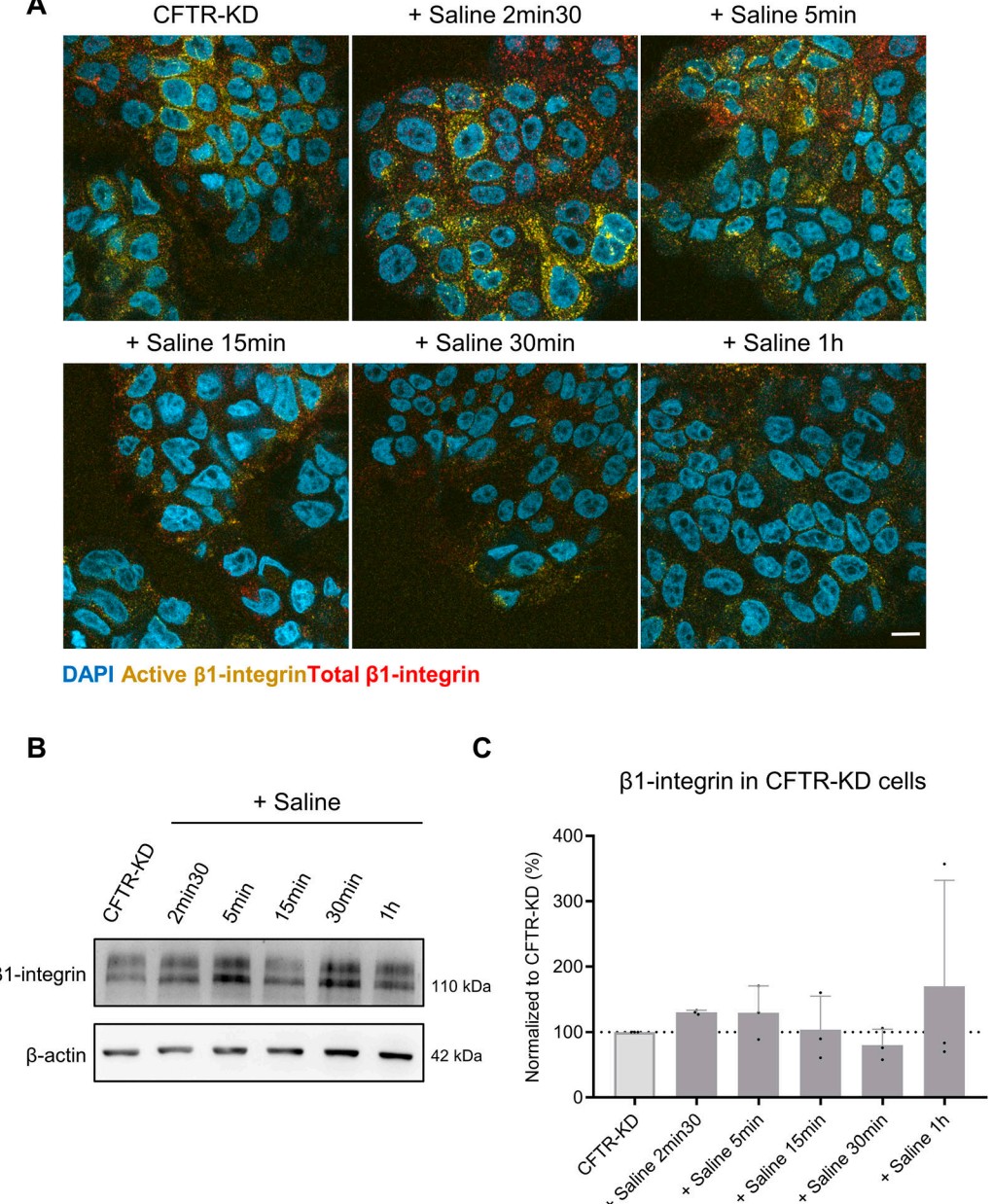

**Figure 2. Kinetics of apical β1-integrin elimination from the CFTR-KD cell surface during rehydration.**
**(A)** Representative apical images from confocal acquisitions of co-immunostaining of both total and active β1-integrins at different time points of saline rehydration (+ Saline 2 min 30 s, 5, 15, 30 min, and 1 h) are shown in polarized CFTR-KD ALI cultures. DAPI: blue; active β1-integrin: yellow; total β1-integrin: red ($N$ = 3). Scale bar: 10 μm. **(B, C)** Representative Western blot and corresponding quantification (C) of total β1-integrin expression at different time points of saline rehydration in CFTR-KD cultures (light gray) and in rehydrated CFTR-KD cells (dark gray). β-Actin was used as a loading control. **(C)** The dotted line in (C) depicts the β1-integrin/β-actin ratio in the CFTR-KD condition ($N$ = 3).
Source data are available for this figure.

## Rehydration of the CFTR-KD Calu-3 cell surface restored YAP1/TAZ protein expression

The contribution of YAP1/TAZ to mechanotransduction is well described (Dupont et al, 2011; Dasgupta & McCollum, 2019). To investigate a potential link between the presence of an ASL and YAP1 expression, we compared YAP1 expression in CFTR-CTL and CFTR-KD

ALI cultures with CFTR-KD cells after rehydration of their apical surface. As shown in Fig 5A, YAP1 expression, as evaluated by immunofluorescence, was strongly decreased in CFTR-KD cells as compared to CFTR-CTL cells. This observation was quantitatively confirmed by Western blot (Fig 5B and C). Next, CTL-ASL from 3-wk-old CFTR-CTL cell cultures was removed and transferred onto the apical side of CFTR-KD cultures for 48 h (CFTR-KD + CTL-ASL).

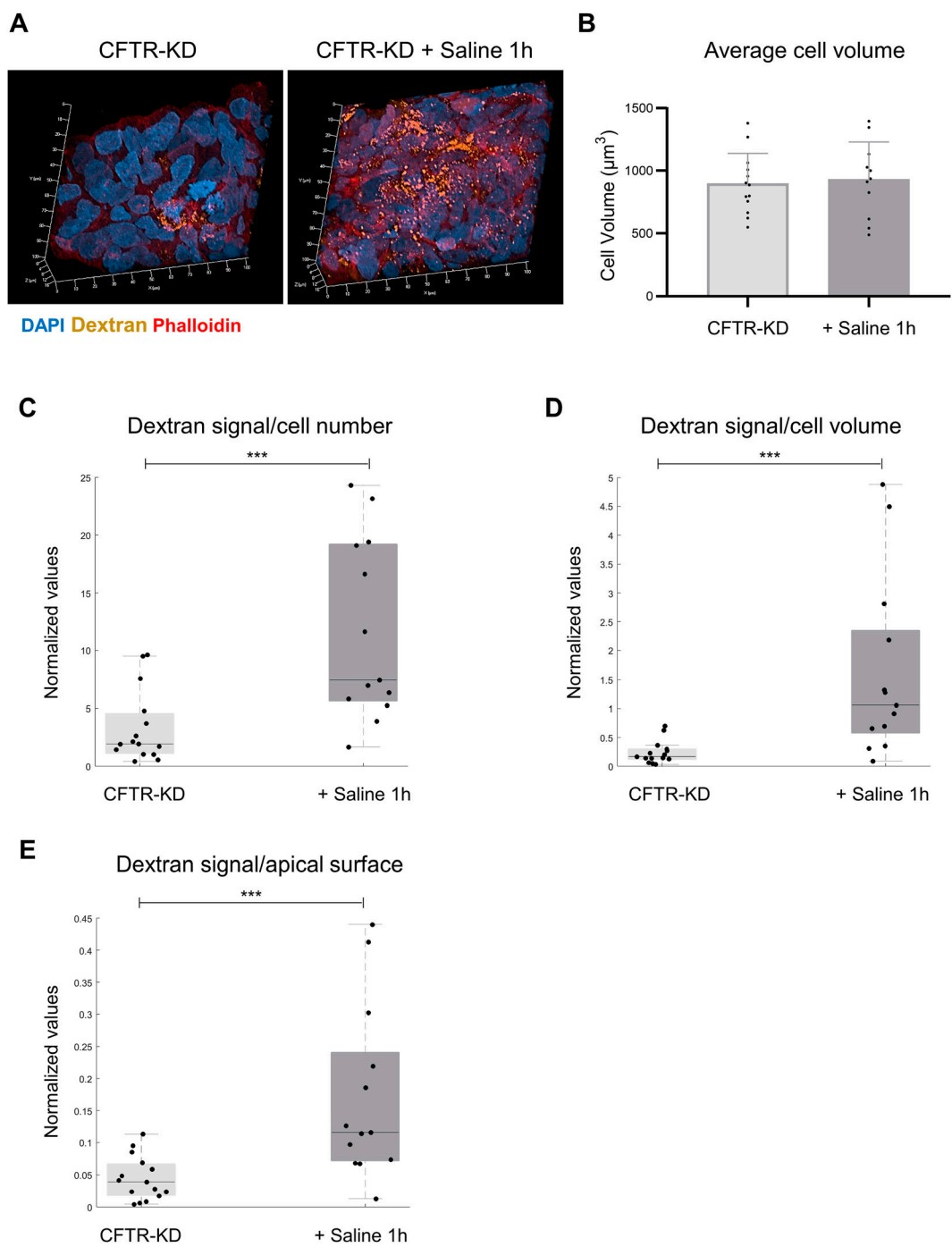

**Figure 3.   Rehydration restores fluid-phase endocytosis in CFTR-KD cells.**
**(A)** Representative images from z-stack confocal acquisitions of dextran–TRITC fluorescence in CFTR-KD cultures and rehydrated polarized CFTR-KD ALI cultures with saline for 1 h (+ Saline 1 h). DAPI: blue; dextran: orange; phalloidin: red. **(B)** Average cell volume ($\mu m^3$) determined from the phalloidin staining was calculated after Imaris reconstruction of confocal images in CFTR-KD (light gray) and CFTR-KD + Saline 1 h (dark gray) conditions ($N$ = 4, $n$ = 2–4). $t$ test, no significance. **(C, D, E)** Median of dextran values before (light gray) and after (dark gray) rehydration of CFTR-KD cultures was determined with Imaris and expressed to the cell number (C), to the average cell volume (D), or to the apical surface (E) ($N$ = 4, $n$ = 3–4). **(C, D, E)** Wilcoxon's rank-sum tests were performed for graphs (C, D, E), $P$ = 1.734 × 10⁻³, $P$ = 3.896 × 10⁻⁴, and $P$ = 7.716 × 10⁻⁴, respectively.
Source data are available for this figure.

In parallel experiments, a similar volume of physiological saline was added to the surface of CFTR-KD cells (CFTR-KD + Saline). Importantly, we observed the re-expression of YAP1 in the cytoplasm, but not in the nucleus, of CFTR-KD cells rehydrated by both means (Fig 5A). The re-expression of YAP1 by ASL or saline rehydration was confirmed by Western blot (Fig 5D and E).

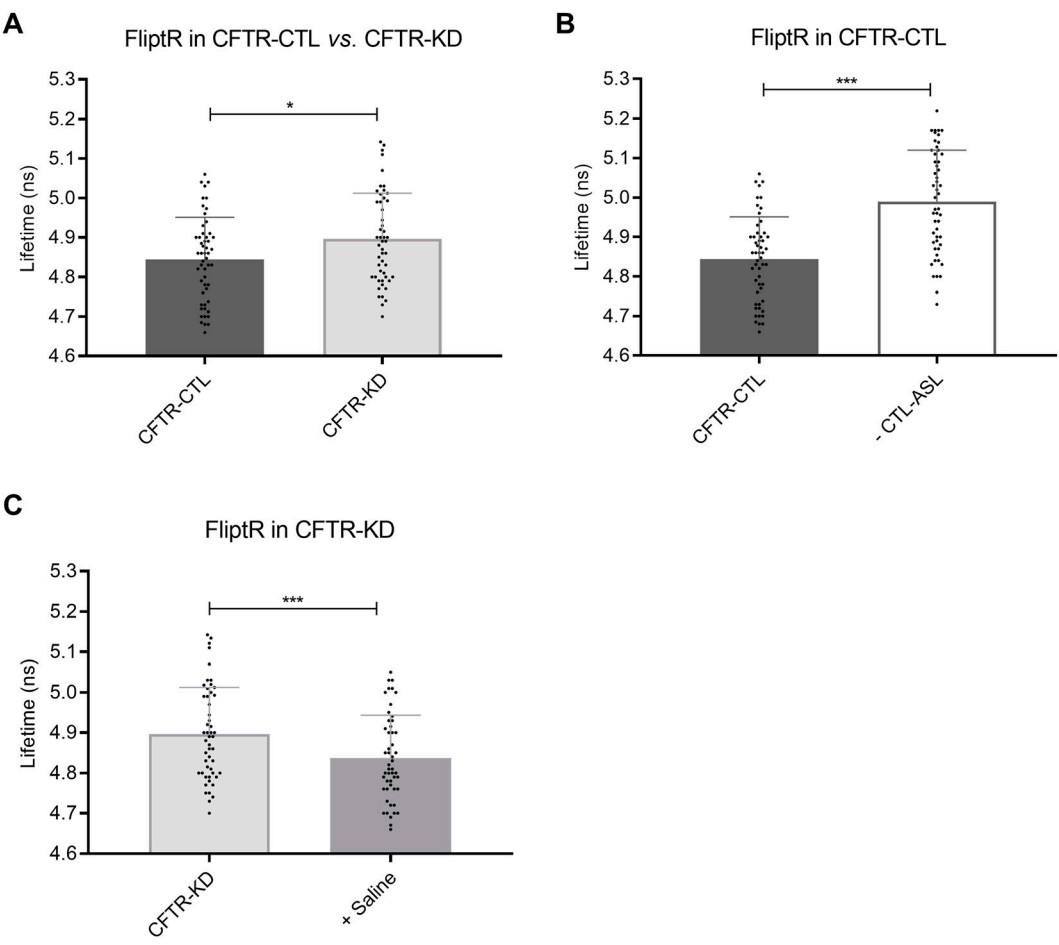

**Figure 4. ASL manipulation modulates membrane tension in CFTR-CTL and CFTR-KD cells.**
Changes in the fluorescence lifetime of the FliptR probe were monitored by FLIM in polarized CFTR-CTL and CFTR-KD ALI cultures to report membrane tension.
**(A)** Steady-state FliptR signal values in CFTR-CTL (black) and CFTR-KD (light gray) cultures. **(B)** FliptR signal values in CFTR-CTL cells before (black) and after (white) ASL removal (−CTL-ASL). **(C)** FliptR signal values in CFTR-KD cells before (light gray) and after (dark gray) the apical addition of 100 μl saline (+ Saline) ($N = 3$, $n > 10$). $P = 0.0179$ (A), $P < 0.0001$ (B), $P < 0.0001$ (C).
Source data are available for this figure.

To determine whether the decreased YAP1 expression in CFTR-KD cells is a direct consequence of *CFTR* knockdown or indirectly results from abnormal polarization of CFTR-KD cells, we compared non-confluent submerged monolayers grown on plastic with ALI cultures on Transwell filters at different time points. YAP1 is mostly localized in nuclei of both CFTR-CTL and CFTR-KD cells when grown as monolayers (Fig S3A). No difference in terms of mRNA (Fig S3B) and protein (Fig S3C and D) expression was observed in both cell lines. We also examined the fraction of YAP1 phosphorylated on S397 (pYAP397), which represents the proteins targeted for proteasomal degradation (Fig S3E and F). Again, no difference in pYAP397 expression was observed between CFTR-CTL and CFTR-KD monolayers. When both cell lines were grown at ALI, we observed an increased expression of YAP1 over time, which reached a stable level in CFTR-CTL cells after 14 d (D14) of culture (Fig S4A and B). Although CFTR-KD cells exhibited a tendency to a higher expression level during the first days of ALI, a marked decrease in YAP1 expression was observed from D11 (Fig S4A and B), which, interestingly, was not caused by a change in YAP1 mRNA

amount (Fig S4C). In addition, mRNA expression decay, as determined by actinomycin D chase experiments, showed that YAP1 mRNA stability of the CFTR-KD cells was not different from that of CFTR-CTL cells (Fig S4D).

Next, we monitored YAP1 protein degradation in both CFTR-CTL and CFTR-KD cells at D16 of ALI culture using the cycloheximide (CHX) chase assay. Quantification of YAP1 expression after 24 h of treatment showed a stronger decrease in CFTR-KD cells as compared to control conditions (Fig 6A and B). We also examined the effects of rehydration with CTL-ASL or saline for 48 h on YAP1 mRNA (Fig S4E) and pYAP397 expression (Fig 6C and D). As shown in Fig 6D, the amount of YAP1 increased and the pYAP397/YAP1 ratio decreased after rehydration of the CFTR-KD apical surface, whereas YAP1 mRNA expression was not changed by this manipulation. Thus, apical rehydration slowed down YAP1 degradation in CFTR-KD cells.

The mRNA expression and protein expression of TAZ, the transcriptional co-activator of the Hippo pathway, were also evaluated in monolayers (Fig S5A–D) and ALI cultures (Figs S5E–G and S6A–E) of CFTR-CTL and CFTR-KD cells. TAZ exhibits a similar pattern of

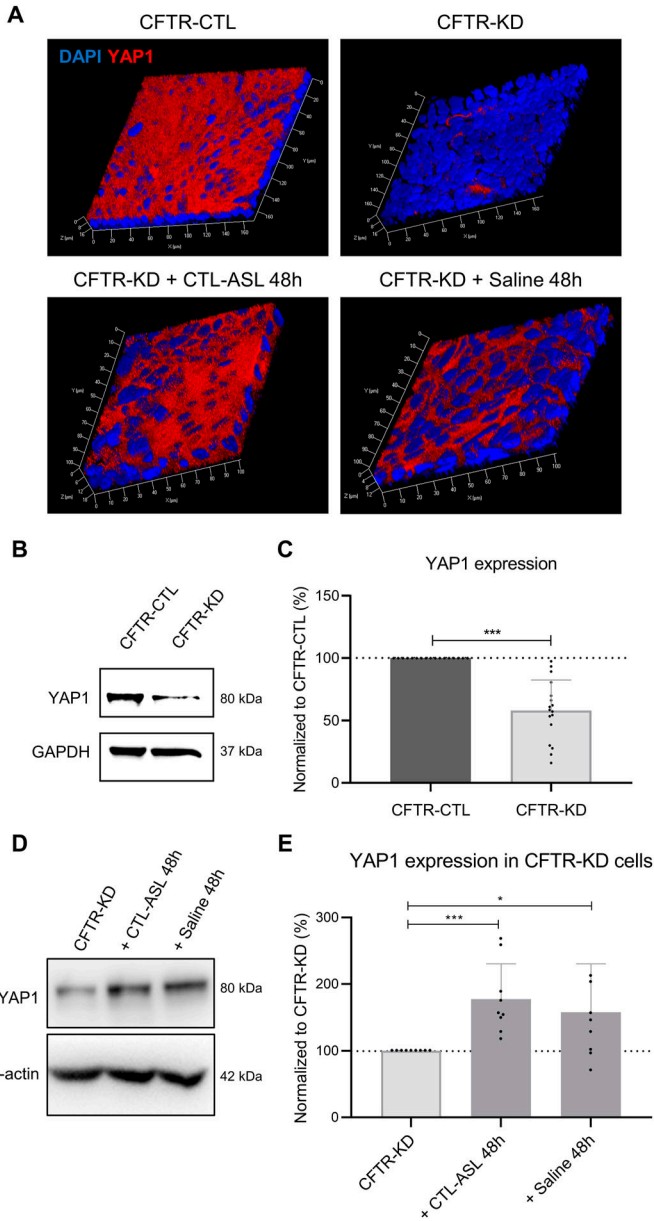

**Figure 5. Rehydration restores YAP1 protein expression in CFTR-KD cells.**
**(A)** Representative images from z-stack confocal acquisitions of immunofluorescence for YAP1 in polarized CFTR-CTL and CFTR-KD ALI cultures, and in CFTR-KD cells rehydrated for 48 h with CTL-ASL (+CTL-ASL 48 h) or saline (+ Saline 48 h). DAPI: blue; YAP1: red ($N = 3$). **(B, C)** Representative Western blot and corresponding quantification (C) of YAP1 expression in CFTR-CTL (black) and CFTR-KD (light gray) cultures. GAPDH was used as a loading control. **(C)** Dotted line in (C) depicts the YAP1/GAPDH ratio in the CFTR-CTL condition ($N = 17$). $P = 4.159 \times 10^{-8}$. **(D, E)** Representative Western blot and corresponding quantification (E) of YAP1 expression in CFTR-KD cultures (light gray) and in rehydrated CFTR-KD cells (dark gray) for 48 h with CTL-ASL (+CTL-ASL 48 h, $P = 4.737 \times 10^{-4}$) or saline (+ Saline 48 h, $P = 1.413 \times 10^{-2}$). β-Actin was used as a loading control. The dotted line in (E) depicts the YAP1/β-actin ratio in the CFTR-KD condition ($N = 9$). Source data are available for this figure.

expression as YAP1 with no differences in mRNA and protein expression in submerged monolayers. In ALI cultures, TAZ expression was decreased in CFTR-KD cells but restored in the cytoplasm by surface rehydration with CTL-ASL or saline for 48 h (Fig S6A–E).

These data suggest that the loss of YAP1 and TAZ expression in late polarized CFTR-KD cells results from decreased stability of the proteins and not from reduced synthesis.

## YAP1 modulates the integrity of airway epithelial junctional complexes

We have previously reported that *CFTR* knockdown was associated with the altered formation of junctional complexes and that apical surface rehydration of CFTR-KD cells restored the expression of tight and adherens junction proteins (Simonin et al, 2022). To examine the contribution of YAP1 to the junctional proteins' network of polarized CFTR-CTL and CFTR-KD cells, we conditionally knocked down *YAP1* by lentiviral transduction of a YAP1 shRNA whose expression is controlled by the IPTG-*LacI* repressor system. As expected, the addition of IPTG for 3–5 d to CFTR-CTL cells polarized at ALI was sufficient to reduce the expression of YAP1 (Fig 7A and E). *YAP1* silencing was associated with the decreased expression of E-cadherin (Fig 7B and E), and claudin-3 and claudin-2 (Fig 7C and E) but not of β-catenin and α1-catenin (Fig 7D and E).

In mirror experiments, we evaluated the effects of apical surface rehydration on junctional proteins of transduced CFTR-KD cells while preventing YAP1 re-expression by the addition of IPTG for 3–5 d. Fig S7 shows representative Western blots for YAP1 (Fig S7A) and junctional proteins (Fig S7B) of CFTR-KD cells treated without or with IPTG for 3–5 d in the absence or presence of rehydrating saline at the apical surface. Consistently with the data obtained after silencing *YAP1* in CFTR-CTL cells, the expression of E-cadherin, claudin-3, and claudin-2 remained low when YAP1 re-expression was prevented during rehydration of the apical surface of CFTR-KD cells (Figs 7F and S7B). Again, the expression of β-catenin and α1-catenin was not affected by *YAP1* silencing (Figs 7F and S7B). As a control, Fig 7G illustrates the re-expression of YAP1 and junctional proteins in rehydrated transduced CFTR-KD cells without the addition of IPTG, reproducing results obtained in original CFTR-KD cells. Of note, IPTG-dependent inhibition of YAP1 expression in transduced CFTR-KD cells did not affect total β1-integrin expression after rehydration, as evaluated by Western blot (Fig S8A and B), and did not re-establish an apical localization of β1-integrin, as shown by immunofluorescence (Fig S8C). These results indicate that YAP1 expression is required to maintain the integrity of the airway epithelium.

## YAP1 degradation in CFTR-KD cells is linked to the apical localization of β1-integrin

The observation that rehydrated CFTR-KD cultures in the presence of IPTG did not re-express β1-integrin at the apical membrane indicates that the ectopic localization of the receptor is YAP1-independent. This prompted us to evaluate whether the presence of the active form of β1-integrin in the apical membrane of dehydrated CFTR-KD cells may transduce intracellular signals to modulate YAP1 expression. To tackle this possibility, we treated CFTR-KD cells with amitriptyline, which is known to trigger β1-integrin internalization (Grassme et al, 2017). As expected, treatment of dehydrated CFTR-KD cultures with amitriptyline for 24–48 h reduced drastically apical β1-integrin expression, as shown by

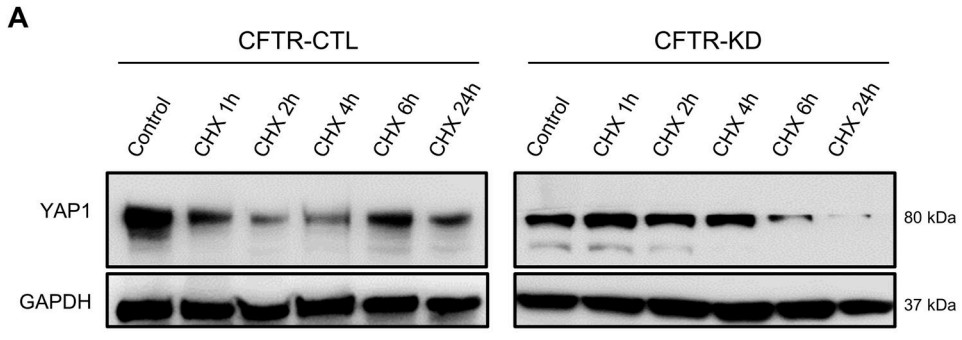

## A

CFTR-CTL                    CFTR-KD

Control, CHX 1h, CHX 2h, CHX 4h, CHX 6h, CHX 24h    Control, CHX 1h, CHX 2h, CHX 4h, CHX 6h, CHX 24h

YAP1                                                          80 kDa

GAPDH                                                        37 kDa

## B

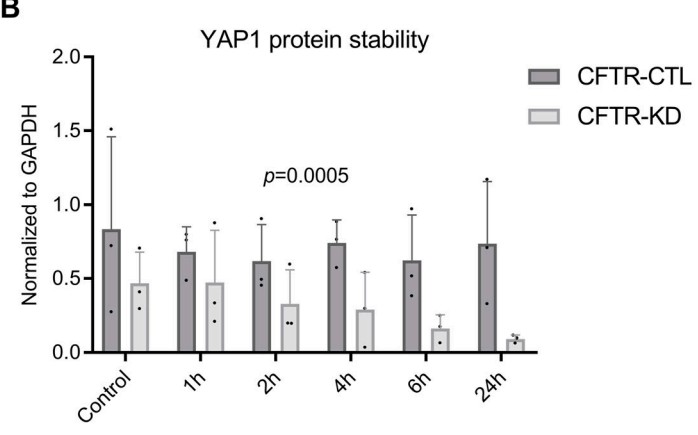

YAP1 protein stability

*p*=0.0005

CFTR-CTL
CFTR-KD

Normalized to GAPDH

Control, 1h, 2h, 4h, 6h, 24h

## C

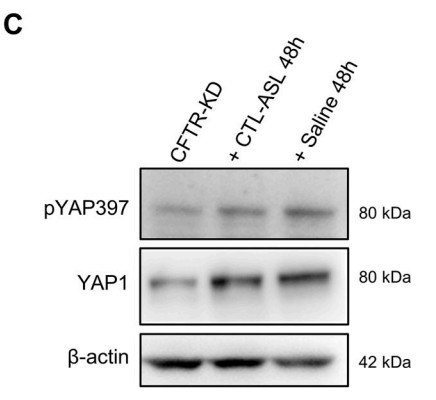

CFTR-KD, + CTL-ASL 48h, + Saline 48h

pYAP397        80 kDa

YAP1            80 kDa

β-actin         42 kDa

## D

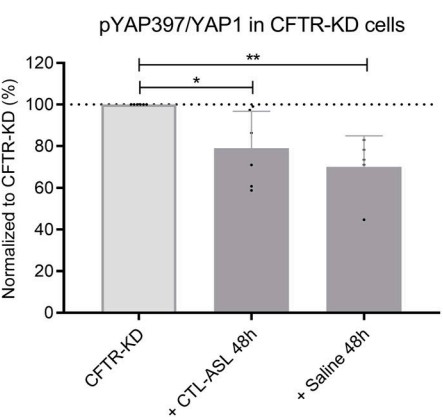

pYAP397/YAP1 in CFTR-KD cells

Normalized to CFTR-KD (%)

*    **

CFTR-KD, + CTL-ASL 48h, + Saline 48h

**Figure 6. Rehydration slows down YAP1 protein degradation in CFTR-KD cells.**
**(A, B)** Representative Western blot and corresponding quantification (B) of YAP1 expression in polarized CFTR-CTL (dark gray) and CFTR-KD (light gray) ALI cultures after treatment with cycloheximide for 1, 2, 4, 6, and 24 h. **(B)** GAPDH was used as a loading control, and data in (B) are expressed as YAP1/GAPDH ratio ($N$ = 3). Two-way ANOVA test, $P$ = 0.0005. **(C, D)** Representative Western blot and corresponding quantification (D) of pYAP397 and YAP1 expression in CFTR-KD cultures (light gray) and in rehydrated CFTR-KD cells (dark gray) for 48 h with CTL-ASL (+CTL-ASL 48 h, $P$ = 0.01) or saline (+ Saline 48 h, $P$ = 0.001). β-Actin was used as a loading control. The dotted line in (D) depicts the pYAP397/YAP1 ratio in the CFTR-KD condition ($N$ = 5–6).
Source data are available for this figure.

immunofluorescence (Fig 8A). Importantly, amitriptyline treatment increased YAP1 protein expression, as evaluated by Western blot (Fig 8B and C) underlining a correlation between the presence of apical β1-integrin and YAP1 protein degradation in dehydrated polarized CFTR-KD cells.

## Discussion

We recently reported that the CF airway epithelium exhibits an apicobasal polarity defect associated with ectopic luminal accumulation of β1-integrin and reduced junctional integrity, thereby promoting *P. aeruginosa* adhesion to the surface and epithelial cytotoxicity (Badaoui et al, 2020, 2023; Simonin et al, 2022). In this study, we show that the Hippo-associated YAP1 effector links apical β1-integrin to the loss of tight and adherens junction proteins.

The apical localization of integrins is thought to result from the redistribution of the protein as opposed to an increased expression, although the mechanisms governing this relocation are still not fully understood. In CF, apical β1-integrin was either observed in airway epithelial cells expressing the commonest

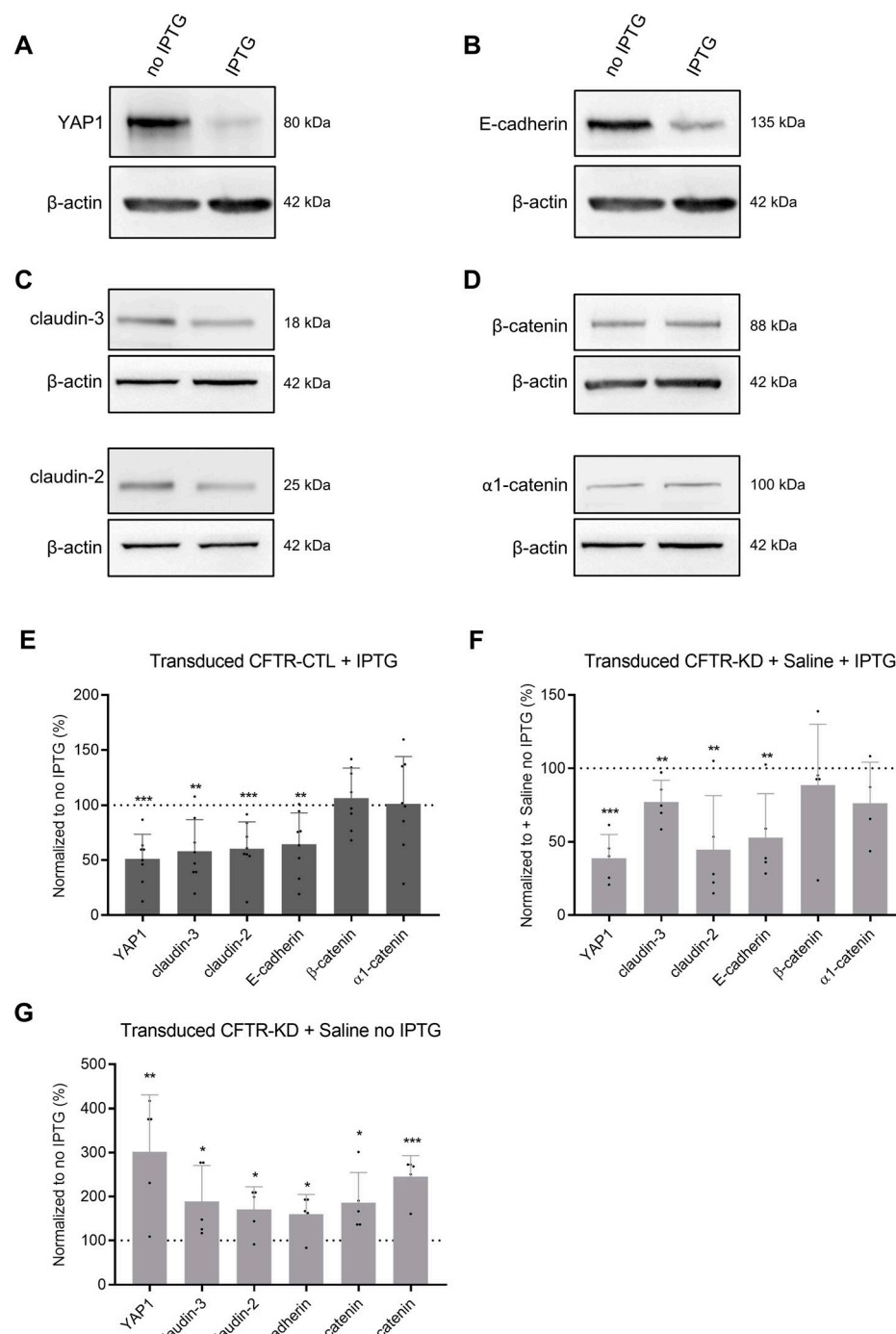

**Figure 7. *YAP1* knockdown and junctional protein expression in transduced CFTR-CTL cells and CFTR-KD cells.**
**(A, B, C, D, E)** Representative Western blots and corresponding quantification (E) of YAP1 (A), E-cadherin (B), claudin-3 and claudin-2 (C), and β-catenin and α1-catenin (D) expression in polarized CFTR-CTL (black) ALI cultures treated or not with IPTG. β-Actin was used as a loading control. Note that YAP1, E-cadherin, and β-actin were immunoblotted from the same membrane, but β-actin blots were duplicated for clarity. The dotted line in (E) depicts the protein/β-actin ratio in the CFTR-CTL, no IPTG condition (N = 8). P = 2.50 × 10⁻⁵ (YAP1), P = 1.04 × 10⁻³ (claudin-3), P = 4.13 × 10⁻⁴ (claudin-2), P = 3.43 × 10⁻³ (E-cadherin). **(F)** Quantification of YAP1, claudin-3, claudin-2, E-cadherin, β1-catenin, and α1-catenin in transduced CFTR-KD cells after rehydration with saline and in the presence of IPTG. The dotted line depicts the protein/β-actin ratio in the rehydrated CFTR-KD cells, no IPTG condition (N = 4–5). P = 3.1 × 10⁻⁵ (YAP1), P = 0.009 (claudin-3), P = 0.010 (claudin-2), P = 0.008 (E-cadherin). **(G)** Quantification of YAP1, claudin-3, claudin-2, E-cadherin, β1-catenin, and α1-catenin in transduced CFTR-KD cells after rehydration with saline and in the absence of IPTG. The dotted line depicts the protein/β-actin ratio in the dehydrated CFTR-KD, no IPTG condition (N = 5). P = 0.008 (YAP1), P = 0.040 (claudin-3), P = 0.016 (claudin-2), P = 0.018 (E-cadherin), P = 0.022 (β-catenin), P = 0.0001 (α1-catenin). Representative Western blots are shown in Fig S6. *YAP1* knockdown prevents its re-expression and those of claudin-3, claudin-2, and E-cadherin, which are normally observed with rehydration.
Source data are available for this figure.

CFTR F508del variant or knocked down for *CFTR* (Grassme et al, 2017; Badaoui et al, 2020). An important function of CFTR is to drive osmotically water movement to the ASL. Because rehydration of the CF epithelial surface rapidly removed apical β1-integrin, it is likely that this phenotype results from the dysfunction or absence of the CFTR channel. Possibly, altered recycling or diminished internalization of β1-integrin may explain the CF phenotype (Bradbury et al, 1992). β1-integrin endocytosis may follow

two routes, namely, a clathrin-dependent or a clathrin- and dynamin-independent pathway (Moreno-Layseca et al, 2021). Using a straightforward approach to monitor fluid-phase endocytosis, our data point to a global defect of endocytosis in CF HAECs.

It was recently reported in fibroblasts that β1-integrin endocytosis is modulated by adjustment of membrane tension in response to mechanical stress (Lolo et al, 2022). Changes in membrane

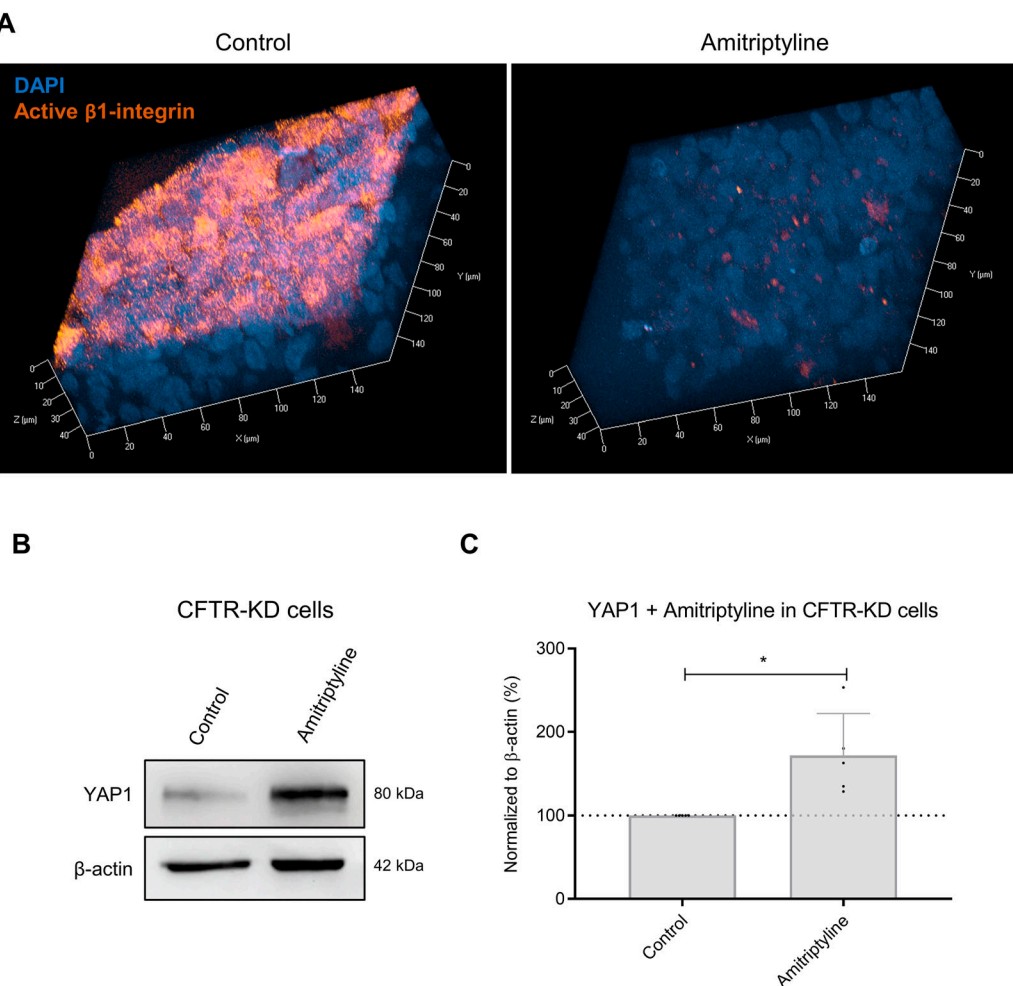

**Figure 8.  Targeting β1-integrin apical localization restored YAP1 expression in CFTR-KD cells.**
**(A)** Representative images from z-stack confocal acquisitions of immunofluorescence for active β1-integrin in polarized CFTR-KD ALI cultures without (control) and after 24–48 h of treatment with amitriptyline. DAPI: blue; active β1-integrin: orange (N = 3). **(B, C)** Representative Western blot and corresponding quantification (C) of YAP1 in response to amitriptyline treatment. β-Actin was used as a loading control. The dotted line in (C) depicts the YAP1/β-actin ratio in the CFTR-KD, control condition (N = 5). P = 0.012.
Source data are available for this figure.

tension directly affect lipid packing, which can be reported by the fluorescent push–pull probe FliptR (Colom et al, 2018). We now show that ASL manipulation in control and CF HAEC cultures affects FliptR lifetime within minutes, making it unlikely that they result from changes in membrane lipid composition. In addition, we found that membrane tension is increased in CF HAECs and that ASL rehydration reduced this tension with kinetics consistent with that of β1-integrin apical disappearance (Moreno-Layseca et al, 2021). A consequence of ASL dehydration would be concentrating the secretome. The crowding of surface and membrane-attached macromolecules may affect membrane shaping, as well as protein–protein and protein–lipid interactions (Kutti Kandy & Radhakrishnan, 2022). For instance, failure of mucociliary clearance in CF is due to abnormal mucus biophysical properties (Okuda et al, 2022). Although the molecular mechanisms remain to be investigated, our data suggest that a dehydrated ASL environment induces a membrane tension increase, which prevents endocytosis.

Both inactive and active forms of β1-integrin were detected at the apical surface of the dehydrated CF airway epithelium. The steady presence of active β1-integrin to the apical membrane is likely to have effects on cell–matrix and cell–cell adhesion dynamics, thereby altering processes such as wound repair or maintenance of epithelial barrier integrity. Increasing evidence suggests that the CF airway epithelium exhibits a partial epithelial–mesenchymal transition phenotype (Amaral et al, 2020). Among typical features of the epithelial–mesenchymal transition, the loss of E-cadherin and claudins (LeSimple et al, 2010; Nilsson et al, 2010; Castellani et al, 2012; Ruan et al, 2014; Simonin et al, 2022) and abnormal regulation of TWIST1 and YAP1 (Quaresma et al, 2020, 2022) were reported in CF HAEC models. It is well established that the Hippo/YAP pathway controls cell proliferation by contact inhibition through cell–cell junctions and epithelial polarity (Gumbiner & Kim, 2014). Homeostasis of the airway epithelium relies thus on the balance between apical signals that inhibit

nuclear YAP1 localization as opposed to basal signals that promote translocation of YAP/TAZ to the nucleus and transcription of proliferation-regulating genes (Elbediwy et al, 2016). In well-polarized HAECs, YAP1 is sequestered in the cytoplasm. Indeed, switching CFTR-CTL cells from monolayers to ALI cultures promoted the relocation of YAP1 from the nucleus to the cytoplasm. Interestingly, *YAP1* silencing after the establishment of apicobasal polarity was associated with the decreased expression of E-cadherin, claudin-3, and claudin-2 in Calu-3 cells, suggesting that cytoplasmic YAP1 is necessary for the maintenance of junctional complexes. We now show that CF HAEC ALI cultures failed to express a sufficient amount of YAP1 to maintain the expression of these junctional proteins. Indeed, although rehydration of CF HAEC cultures rescued E-cadherin, claudin-3, and claudin-2 expression (Simonin et al, 2022), this correction was fully prevented by *YAP1* knockdown. Thus, YAP1 expression is critical in maintaining the integrity of airway epithelial junctional complexes and points to YAP1 degradation as a mediator of the altered structural integrity of the CF airway epithelium. This conclusion, however, is in apparent contradiction with a recent study by Quaresma and collaborators that reported that YAP1 is aberrantly active in F508del-CFTR–expressing cells (Quaresma et al, 2022). Although the different outcomes may be due to the cell models and/or the experimental conditions used, that is, CFBE or 16HBE cells expressing wild-type or F508del-CFTR grown under submerged conditions as compared to CFTR-CTL and CFTR-KD Calu-3 cells polarized at ALI, the two studies indicate that Hippo/YAP signaling is dysregulated in CF HAECs. It is tempting to propose that abnormal F508del-CFTR/YAP1 interaction may occur in basal progenitor cells, whereas cytoplasmic YAP1 is degraded in luminal cells, thereby influencing the repair process after injury toward a hyper-proliferating and partially differentiated airway epithelium (Amaral et al, 2020). It is worth noting that each cell type that constitutes the native airway epithelium may express specific sets of ion channels, transporters, and membrane receptors that would contribute to the ASL volume regulation. Thus, CFTR dysfunction in primary airway epithelial cells may have a more dynamic effect on the YAP1–junctional protein relationship depending on the environmental conditions.

We finally report that apical $\beta$1-integrin accumulation and YAP1 degradation in dehydrated CF HAEC cultures are correlated events. Apical $\beta$1-integrin is thought to be trapped in ceramide-rich regions (Grassme et al, 2017), which in CF results from the imbalanced activities of acid ceramidase and acid sphingomyelinase (Teichgraber et al, 2008). Thus, inhibition of acid sphingomyelinase with amitriptyline normalized the subcellular distribution pattern of $\beta$1-integrin (Grassme et al, 2017). Interestingly, amitriptyline rescued YAP1 expression in dehydrated CF HAEC cultures. Among several possibilities, apical $\beta$1-integrin may overactivate Hippo/YAP signaling, possibly through YAP1 ubiquitination, ultimately leading to YAP1 degradation. Alternatively, apical $\beta$1-integrin may promote YAP1 degradation independently of the Hippo core kinase cascade, possibly through actin cytoskeleton remodeling.

In summary, by manipulating $\beta$1-integrin localization, analyzing downstream Hippo pathway components, and exploring the functional consequences of modulating these interactions in CF HAEC models, we have unveiled a correlation between the apical $\beta$1-integrin and Hippo signaling in CF airway epithelial cells. Our results also highlight the essential role of the presence of an ASL volume in the maintenance of barrier integrity of the airway epithelium. They are in agreement with a chronological cascade of events mediated through apical rehydration of CFTR-KD cultures, triggering apical $\beta$1-integrin disappearance, YAP1 rescue, and thus re-establishment of junctional proteins. Altogether, this study provides new light for understanding cell behavior in lung tissue homeostasis and disease processes and supports therapeutic strategies to restore CFTR activity and ASL hydration in people with CF.

# Materials and Methods

Key reagents and resources used in this study are listed in Table S1.

### Cell culture and treatments

CFTR-CTL and CFTR-KD cells expressing the wild-type CFTR and knocked down for the *CFTR* gene, respectively, were generated by CRISPR/Cas9 from Calu-3 cells, a non–small-cell lung cancer epithelial cell line, as previously reported (Bellec et al, 2015). Cells were maintained in culture as detailed elsewhere (Simonin et al, 2022). Well-polarized CFTR-CTL and CFTR-KD Calu-3 epithelia were established by seeding $1.75 \times 10^5$ cells onto 0.33-cm$^2$ porous (0.4 $\mu$m) Transwell polyester inserts (Transwell 3470; Corning Life Sciences) and cultured for 5 d under submerged conditions until confluence. Epithelial polarization was optimized by culturing cells at the ALI for 15 d.

150 $\mu$g/ml CHX (Sigma-Aldrich) prepared in DMSO was added to the basal side of polarized CFTR-CTL and CFTR-KD cultures for 1, 2, 4, 6, and 24 h at 37°C. CHX-treated cells were compared with the control condition (24 h, DMSO). Similarly, 30 $\mu$M amitriptyline (Sigma-Aldrich) prepared in water was added to the basal side for 24–48 h at 37°C and treated cultures were compared with the control condition (24–48 h, water).

### Cell transduction

Custom lentiviral particles expressing an inducible shRNA targeting the 3'UTR non-coding region of *YAP1* (clone sequence: CCCAGTTAAATGTTCACCAAT; reference: TRCN0000107265) cloned into the pLKO-puro-IPTG-3xLacO vector were purchased from Merck. Proliferating CFTR-CTL and CFTR-KD cells were transduced at a multiplicity of infection of 2 in the presence of 8 $\mu$g/ml hexadimethrine bromide. 4 d after the transduction, transduced cells were selected for 12 d with 5 $\mu$g/ml puromycin before seeding on Transwell filters and grown at ALI. *YAP1* silencing was induced by adding 100 $\mu$M of IPTG (isopropyl-$\beta$-D-thiogalactopyranoside) to the basal side of the filters. IPTG was renewed every day and maintained for 3–5 d.

## Apical surface liquid manipulation

Polarized CFTR-CTL epithelia were maintained with their natural ASL (CTL-ASL), which was carefully removed for some experiments (−CTL-ASL), and the volume was measured with a micropipette. In parallel, we exposed polarized CFTR-KD epithelia to different rehydrating conditions by adding CTL-ASL (+CTL-ASL) collected from one CFTR-CTL culture or by adding to the apical side the same volume of a physiological saline (+ saline) with the same osmolarity, containing NaCl (154 mM), Hepes (10 mM), and $CaCl_2$ (1.2 mM) for the amount of time indicated in the text. No apical washing was performed after CTL-ASL removal or before CTL-ASL versus saline addition. As reported previously, rehydration of CFTR-KD cultures with either CFTR-ASL or saline restored the expression of junctional proteins (Simonin et al, 2022).

## Western blot

Protein extraction and Western blotting were performed as detailed previously (Simonin et al, 2022). All antibodies and their sources are listed in Table S1.

## Immunofluorescence and confocal microscopy

CFTR-CTL and CFTR-KD cells were fixed using 4% PFA (158127; Sigma-Aldrich) for 15 min at RT and permeabilized for 15 min at RT with 0.2% Triton X-100 (T-8787; Sigma-Aldrich) buffer. The non-specific sites were blocked with Superblock solution (37580; Thermo Fisher Scientific) for 15 min at RT, and samples were then incubated overnight with primary antibodies (Table S1) at 4°C. After 3 × 5 min PBS washing, secondary antibodies were applied for 1 h at RT, whereas DAPI (Cat. No. A4099; AppliChem) was used for nuclear counterstaining and phalloidin-iFluor 647 (ab176459; Abcam) was used for the subcortical actin staining. The fluorescence images and Z-stacks were acquired with a LSM700 confocal microscope and ZEN software (ZEISS). The images were analyzed using ZEN and ImageJ software.

## Fluid-phase endocytosis

Fluid-phase endocytosis was evaluated in polarized CFTR-KD cultures rehydrated with saline for 1 h (+saline 1 h) using the dextran–TRITC (tetramethylrhodamine) probe (D1868; Molecular Probes). Both dehydrated and rehydrated cultures were washed with an endocytosis medium (cell culture medium supplemented with 20 mM Hepes and 0.2% BSA) and incubated with 50 $\mu$l of this medium for 45 min at 4°C to inhibit the endocytosis processes. The apical medium was then replaced with the endocytosis medium containing 3 mg/ml dextran–TRITC and incubated for 2 min 30 s at 37°C to resume endocytosis. At the end of the incubation period, cells were well washed with cold PBS/0.2% BSA, fixed with 4% PFA for 15 min, and permeabilized for 45 min at RT with 0.25% Tween (A4974; PanReac AppliChem). After 3 × 5 min PBS washing, DAPI was used for nuclear counterstaining and phalloidin-iFluor 647 for the subcortical actin staining. The fluorescence images and Z-stacks were acquired with a LSM800

confocal microscope and ZEN software (averaging 4x and 16 bits per pixel). The images were analyzed using ZEN and Imaris software. 3D z-stack confocal images were converted with the Imaris File converter 9.9.1.

The membrane architecture was reconstructed with Imaris, based on the phalloidin subcortical actin staining. Each dextran–TRITC signal was defined as a spot (1 $\mu$m xy, 2 $\mu$m z, and quality 300). Next, the shortest distance between the membrane and each dextran spot was determined using both Imaris and MATLAB software. The value 0 corresponds to the dextran spot restricted to the membrane. The number of dextran spots per cell was normalized to the DAPI staining from confocal images, which allows for determining the number of nuclei with CellPose software. For normalization of dextran spots to cell volume, the entire volume of the reconstructed images was determined and then averaged by the number of nuclei. Finally, membrane reconstruction was used to determine the apical surface for normalization of the dextran spots.

## Membrane tension

Membrane tension was evaluated by FLIM of polarized CFTR-CTL and CFTR-KD cultures incubated for 1 h at the basal side with the FliptR fluorescent probe (1 $\mu$l/ml, SC020; Spirochrome) under controlled temperature and atmosphere (37°C, 5% $CO_2$). FLIM imaging was performed using a Nikon Eclipse Ti A1R microscope equipped with a Time-Correlated Single-Photon Counting module from PicoQuant, as previously described (Colom et al, 2018), and a water immersion Apo LWD 40X/1.15 N.A. objective (Nikon). During acquisitions, cells were maintained at 37°C and 5% $CO_2$ with a micro-incubator (Okolab). FliptR fluorescence lifetime was determined for at least 10 positions, always at the same height within each Transwell filter. SymPhoTime 64 software (PicoQuant) was used to fit fluorescence decay data (from full images or regions of interest) to a dual exponential model after deconvolution for the instrument response function (measured using the backscattered emission light of a 1 $\mu$M fluorescein solution with 4M KI). Data were expressed as the mean ± SD of the mean.

## RNA extraction and qRT-PCR

Total RNA was extracted from CFTR-CTL and CFTR-KD cells with RNeasy Mini Kit (Cat. No.74106; QIAGEN) and prepared for qRT-PCR as detailed elsewhere (Simonin et al, 2022). Primer pairs (Microsynth) used are shown in Table S2. mRNA expression is represented as the absolute value ($2^{-\Delta Ct}$) normalized to GAPDH or 18S expression.

## mRNA stability assay

mRNA stability was evaluated in polarized CFTR-CTL and CFTR-KD cells after the addition of 5 $\mu$g/ml actinomycin D in the basal side for 4 and 8 h at 37°C. Treated cells were compared with the control condition (8 h, DMSO). Next, cells were washed with PBS, and total RNA was extracted using RNeasy Mini Kit (QIAGEN). YAP1 and TAZ mRNA stability was determined by qRT–PCR.

## Statistical analysis

Values were represented as the mean ± SEM unless indicated differently in the text. Statistical tests were conducted using Sig-maStat (Systat Software, Inc.) or GraphPad Prism software. The difference between the two groups was analyzed by the $t$ test, whereas the difference between more than two groups was tested using the two-way analysis of variance (ANOVA) followed by Holm–Sidak post hoc tests, unless indicated differently in the text. $P < 0.05$, $P < 0.01$, and $P < 0.001$ were considered significant and represented as *, **, and ***, respectively. MATLAB software was used for the statistical analysis of dextran–TRITC endocytosis. $N$ defines the number of performed experiments, and $n$ defines the number of technical replicates for each experiment.

# Data Availability

The dataset for this article can be found at https://doi.org/10.26037/yareta:6tlnqytvq5gtlljgngfu7bjwv4.

# Supplementary Information

# Acknowledgements

This work was supported by a grant from the Swiss National Science Foundation (310030_204167/1). We would like to thank the Faculty of Medicine, University of Geneva Core Bioimaging and READS Facilities. We also thank Dr. Rana El Masri for her comments on the article.

## Author Contributions

JL Simonin: conceptualization, formal analysis, investigation, methodology, and writing—original draft, review, and editing.
C Tomba: conceptualization, formal analysis, investigation, methodology, and writing—review and editing.
V Mercier: conceptualization, formal analysis, investigation, methodology, and writing—review and editing.
M Bacchetta: formal analysis and methodology.
T Idris: formal analysis, methodology, and writing—review and editing.
M Badaoui: formal analysis, validation, investigation, methodology, and writing—review and editing.
A Roux: conceptualization, resources, methodology, and writing—review and editing.
M Chanson: conceptualization, resources, data curation, supervision, funding acquisition, validation, project administration, and writing—original draft, review, and editing.

## Conflict of Interest Statement

The authors declare that they have no conflict of interest.

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
