## [Reviewer comments · Life Science Alliance]

Life Science Alliance

Apical dehydration impairs cystic fibrosis airway epithelium barrier via a β 1-integrin/YAP1 pathway

Juliette L. Simonin, Caterina Tomba, Vincent Mercier, Marc Bacchetta, Tahir Idris, Mehdi Badaoui, Aurélien Roux, and Marc Chanson

DOI: <https://doi.org/10.26508/lsa.202302449>

Corresponding author(s): Marc Chanson, University of Geneva

Review Timeline:

Submission Date:	2023-10-19
Editorial Decision:	2023-12-11
Revision Received:	2024-01-24
Editorial Decision:	2024-01-25
Revision Received:	2024-01-31
Accepted:	2024-01-31

Transaction Report:

December 11, 2023

Re: Life Science Alliance manuscript #LSA-2023-02449-T

Prof. Marc Chanson
University of Geneva
PHYM
Rue Michel-Servet, 1
Geneva 1211
Switzerland

Dear Dr. Chanson,

Thank you for submitting your manuscript entitled "Apical dehydration impaired cystic fibrosis airway epithelium integrity via a β 1-integrin/YAP1 mechanism" to Life Science Alliance. The manuscript was assessed by expert reviewers, whose comments are appended to this letter. We invite you to submit a revised manuscript addressing the Reviewer comments.

Thank you for this interesting contribution to Life Science Alliance. We are looking forward to receiving your revised manuscript.

Sincerely,

B. MANUSCRIPT ORGANIZATION AND FORMATTING:

Reviewer #1 (Comments to the Authors (Required)):

This is a very interesting manuscript that reports important findings about changes in the plasma membrane tension of HAEC with CF. The authors identify a few parameters of interest, in particular, the role of YAP1 signaling in maintaining junction proteins.

A few minor points still need to be addressed.

In Fig2, the authors report the apical expression of B1-integrin and then its rapid disappearance upon the addition of ASL from control cells or saline.

To be convinced of the changes in the apical localization of the protein, it would take a co-localization with an apical marker. Alternatively, a cross-section view of the cells would enable the reader to see the change.

The difference between total and active integrin is unclear and should be better defined.

There are several instances in the paper where the authors mention that changes in plasma membrane tension result in changes in integrin endocytosis. This is an overstatement. General changes in endocytosis have been measured with a dye, but not the endocytosis of beta integrin directly.

In the methods section, more details are required about the ASL that is collected from control cells and transferred to CFTR-KD cells. What method is used to collect the ASL? are cell debris removed? how do you control the composition and volume?

Finally, in the last paragraph of the discussion. This reviewer does not think that you have established a direct link between apical b-integrin and the Hippo signaling, but rather a correlation.

Reviewer #2 (Comments to the Authors (Required)):

The manuscript by Simonin et al. entitled "Apical dehydration impaired cystic fibrosis airway epithelium integrity via a β 1-integrin/YAP1 mechanism" investigates the mechanisms linking ASL to epithelial barrier integrity. The results of the experiments conducted in this study suggest that that dehydration of the CF ASL affects epithelial plasma membrane tension, resulting in ectopic activation of a β 1-integrin/YAP1 signaling that is associated with degradation of junctional proteins.

This is a well-written manuscript on a crucial, but not well-understood topic, that offers interesting results and conclusions. The methods utilized are adequate, the experiments well-described and the results convincing. Conclusions provided in the manuscript are supported by the data.

Comments:

ASL dehydration in CF is associated with an increase in mucin concentration. This should be mentioned and discussed.

While Calu-3 cells are a good model, primary human HBE cells exhibit a larger variety of cell types and this may affect responses to alterations in ASL. The consequences of having various epithelial cell types (also as it relates to the expression of implicated proteins studied here) may be discussed.

A cartoon that summarizes the findings would be a great addition for understanding the discovered concepts.

In Figure 1: a quantitation of total (A) and active (B) integrin staining would be of advantage.

Figure 1: Please explain abbreviation NT (non-treated) in figure legend.

Answers to Reviewers' comments:

Answers to Reviewers' comments are in **bold**.

Reviewer #1:

This is a very interesting manuscript that reports important findings about changes in the plasma membrane tension of HAEC with CF. The authors identify a few parameters of interest, in particular, the role of YAP1 signaling in maintaining junction proteins.

A few minor points still need to be addressed.

We thank the Reviewer for her/his interest in our work.

1. In Fig2, the authors report the apical expression of $\beta 1$ -integrin and then its rapid disappearance upon the addition of ASL from control cells or saline.

To be convinced of the changes in the apical localization of the protein, it would take a co-localization with an apical maker. Alternatively, a cross-section view of the cells would enable the reader to see the change.

The difference between total and active integrin is unclear and should be better defined.

We thank the Reviewer for his comments on Figure 2. Active from total $\beta 1$ -integrin can be distinguished using the 9EG7 antibody, which binds to an epitope accessible only after activation of the protein. This is now defined in the revised manuscript (p.7, l.120-122) and a reference has been added (Lenter et al, 1993).

To convince about the changes in the apical localization of $\beta 1$ -integrin, we now show 3D reconstruction of z-stack images of the immunostained CFTR-KD cultures before and after rehydration, confirming the disappearance of the apical signal. This has led to changes in the main text of the "Results" section (p.8, l.136-137) as well as in the addition of a new supplemental Figure (Figure S1) and Figure legend (p.31, l.735-740).

2. There are several instances in the paper where the authors mention that changes in plasma membrane tension result in changes in integrin endocytosis. This is an overstatement. General changes in endocytosis have been measured with a dye, but not the endocytosis of beta integrin directly.

We agree with the remark of the Reviewer. We have rephrased the text where appropriate (abstract p.3, l.44-46; headline p.8, l.142; p.9, l.180; p.15, l.308; p.17, l.375-376).

3. In the methods section, more details are required about the ASL that is collected from control cells and transferred to CFTR-KD cells. What method is used to collect the ASL? are cell debris removed? How do you control the composition and volume?

We have previously reported how manipulation of the ASL was performed (Simonin et al., 2022). More details are now provided in the "Materials and Methods" section of the revised manuscript (p.19, l.415-416 and l.420-423).

4. Finally, in the last paragraph of the discussion. This reviewer does not think that you have established a direct link between apical β -integrin and the Hippo signaling, but rather a correlation.

We have rewritten the sentence as follows: "we have unveiled correlation between apical $\beta 1$ -integrin and Hippo signaling in CF airway epithelial cells" (p.17, l.371).

Reviewer #2:

The manuscript by Simonin et al. entitled "Apical dehydration impaired cystic fibrosis airway epithelium integrity via a β 1-integrin/YAP1 mechanism" investigates the mechanisms linking ASL to epithelial barrier integrity. The results of the experiments conducted in this study suggest that dehydration of the CF ASL affects epithelial plasma membrane tension, resulting in ectopic activation of a β 1-integrin/YAP1 signaling that is associated with degradation of junctional proteins.

This is a well-written manuscript on a crucial, but not well-understood topic that offers interesting results and conclusions. The methods utilized are adequate, the experiments well-described and the results convincing. Conclusions provided in the manuscript are supported by the data.

We thank the Reviewer for her/his interest in our work.

1 ASL dehydration in CF is associated with an increase in mucin concentration. This should be mentioned and discussed.

We have extended our discussion (p.15, 1.308-314) to introduce the concept of macromolecules crowding due to loss of ASL volume in CF and its consequence on biophysical properties of mucins as well on protein-protein and protein-lipid interactions.

2. While Calu-3 cells are a good model, primary human HBE cells exhibit a larger variety of cell types and this may affect responses to alterations in ASL. The consequences of having various epithelial cell types (also as it relates to the expression of implicated proteins studied here) may be discussed.

We thank the Reviewer for this important comment. Whereas Calu-3 cells are a good model to study the relationship between CFTR function and ASL volume, they are not representative of the primary airway epithelium, which is constituted of a variety of basal and surface epithelial cells. Each cell type may express specific sets of ion channels, transporters and membrane receptors that would contribute to the final ASL volume regulation. Thus, CFTR dysfunction in primary airway epithelial cells may have a more dynamic effect on YAP1-junctional proteins relationship depending of the environmental conditions. This is now indicated in the "Discussion" section of the revised manuscript (p.16, 1.353-p.17, 1.357).

3. A cartoon that summarizes the findings would be a great addition for understanding the discovered concepts.

Thank you for the suggestion. We have included to the revised manuscript a Graphical abstract that summarizes our findings.

4. In Figure 1: a quantitation of total (A) and active (B) integrin staining would be of advantage.

We have added quantitation of total and active integrin staining in new panels C (total) and D (active) of Figure 1. This has led to changes in the main text of the "Results" section (p.7, 1.130-131) and in the legend of Figure 1 (p.27, 1.641-645).

5. Figure 1: Please explain abbreviation NT (non-treated) in figure legend.

Thank you. This is now indicated in the legend of Figure 1 (p.27, 1.639-640).

January 25, 2024

RE: Life Science Alliance Manuscript #LSA-2023-02449-TR

Prof. Marc Chanson
University of Geneva
Cell Physiology and Metabolism
Rue Michel-Servet, 1
Geneva 1211
Switzerland

Dear Dr. Chanson,

Thank you for submitting your revised manuscript entitled "Apical dehydration impaired CF airway epithelium integrity via a β 1-integrin/YAP1 mechanism". We would be happy to publish your paper in Life Science Alliance pending final revisions necessary to meet our formatting guidelines.

- please be sure that the authorship listing and order is correct
- please add a Category for your manuscript in our system
- please add the Twitter handle of your host institute/organization as well as your own or/and one of the authors in our system
- please note that titles in the system and the manuscript file must match
- please use the [10 author names et al.] format in your references (i.e., limit the author names to the first 10)

Figure Checks:

- please add sizes next to all blots
- the link in your Cover Letter for the source data files does not work. These Source Data figures should be uploaded with your manuscript. One Source Data figure per manuscript figure. Source Data will be critical to explain why the beta-actin blots are identical in Figure 7A and B. Also why there are duplicate beta-actin blots used throughout Figure S7A and B. This should be made clear in the respective figure legends as well.

A. FINAL FILES:

B. MANUSCRIPT ORGANIZATION AND FORMATTING:

Sincerely,

January 31, 2024

RE: Life Science Alliance Manuscript #LSA-2023-02449-TRR

Prof. Marc Chanson
University of Geneva
Cell Physiology and Metabolism
Rue Michel-Servet, 1
Geneva 1211
Switzerland

Dear Dr. Chanson,

Thank you for submitting your Research Article entitled "Apical dehydration impairs cystic fibrosis airway epithelium barrier via a β 1-integrin/YAP1 pathway". It is a pleasure to let you know that your manuscript is now accepted for publication in Life Science Alliance. Congratulations on this interesting work.

DISTRIBUTION OF MATERIALS:

Again, congratulations on a very nice paper. I hope you found the review process to be constructive and are pleased with how the manuscript was handled editorially. We look forward to future exciting submissions from your lab.

Sincerely,
